# Edge-Rich Interconnected Graphene Mesh Electrode with High Electrochemical Reactivity Applicable for Glucose Detection

**DOI:** 10.3390/nano11020511

**Published:** 2021-02-17

**Authors:** Van Viet Tran, Duc Dung Nguyen, Mario Hofmann, Ya-Ping Hsieh, Hung-Chih Kan, Chia-Chen Hsu

**Affiliations:** 1Department of Physics, National Chung Cheng University, Chiayi 621, Taiwan; viettran.apc@gmail.com (V.V.T.); ddnguyen161@gmail.com (D.D.N.); phyhck@ccu.edu.tw (H.-C.K.); 2Center for High Technology Development, Vietnam Academy of Science and Technology, Hanoi 100000, Vietnam; 3Department of Physics, National Taiwan University, Taipei 10617, Taiwan; mario@phys.ntu.edu.tw; 4Institute of Atomic and Molecular Sciences, Academia Sinica, Taipei 106, Taiwan; yphsieh@gate.sinica.edu.tw

**Keywords:** nickel mesh, graphene mesh, electrochemical reactivity, electron transfer rate, glucose detection, low vacuum annealing

## Abstract

The development of graphene structures with controlled edges is greatly desired for understanding heterogeneous electrochemical (EC) transfer and boosting EC applications of graphene-based electrodes. We herein report a facile, scalable, and robust method to produce graphene mesh (GM) electrodes with tailorable edge lengths. Specifically, the GMs were fabricated at 850 °C under a vacuum level of 0.6 Pa using catalytic nickel templates obtained based on a crack lithography. As the edge lengths of the GM electrodes increased from 5.48 to 24.04 m, their electron transfer rates linearly increased from 0.08 to 0.16 cm∙s^−1^, which are considerably greater than that (0.056 ± 0.007 cm∙s^−1^) of basal graphene structures (defined as zero edge length electrodes). To illustrate the EC sensing potentiality of the GM, a high-sensitivity glucose detection was conducted on the graphene/Ni hybrid mesh with the longest edge length. At a detection potential of 0.6 V, the edge-rich graphene/Ni hybrid mesh sensor exhibited a wide linear response range from 10.0 μM to 2.5 mM with a limit of detection of 1.8 μM and a high sensitivity of 1118.9 μA∙mM^−1^∙cm^−2^. Our findings suggest that edge-rich GMs can be valuable platforms in various graphene applications such as graphene-based EC sensors with controlled and improved performance.

## 1. Introduction

Graphene possesses unique properties, including excellent transparency, high thermal and electrical conductivity, large specific surface area, and excellent electrochemical (EC) stability which make it one of the most attractive materials [1,2,3,4]. Graphene has been utilized in various applications such as transparent conducting electrodes [5], energy devices [6,7], electronic and optoelectronic devices [8,9], and bio- and chemical-sensing devices [10,11]. In general, EC performance of an electrode is determined by its electron transfer kinetics and the interactions between molecules and electrode surfaces [12,13]. Recently, Yuan et al. reported that graphene edges exhibited higher specific capacitance, faster electron transfer rate, and stronger electrocatalytic activity than those of graphene basal plane [14]. Furthermore, Li et al. reported that a nanometer-thick linear graphene edge nanoelectrode showed much better EC performance compared with traditional carbon fiber microelectrodes [15]. Moreover, Pavlov et al. used theories and computations to show that the graphene edges provide ca. 4 times faster electron transfer in comparison with the basal plane [16]. Therefore, edge-rich graphene, due to its prevalent defects and dangling bonds, is considered an ideal electrode material for energy storage [17,18] and bio- and chemical-sensing applications [19,20,21].

Owing to their superior large periphery (compared to basal/planar structure), graphene mesh (GM) architectures, comprising of interconnected graphene wires, are believed as potent edge-rich graphene electrodes for advanced EC applications. Several methods have been employed to fabricate GM, including electron beam lithography [22], nanoimprint lithography [23], nanosphere-assisted lithography [24,25], template-assisted lithography [26,27], and self-organized growth [28]. However, most of these methods require complex lithography and etching (such as reactive ion etching or toxic etching solution) processes, which are costly and hard to scale up. Moreover, chemical contamination is another obstacle in the existing fabrication processes, which may cause degradation in performance of GM [29]. Recently, we developed a facile method to fabricate nickel (Ni) meshes covered with conformal graphene coating by the combination of crack lithography and rapid thermal annealing techniques [30]. Our fabrication method has advantages of simplicity, robustness, low cost, scalability, and minimal chemical contamination. Inspired by these earlier findings, we hypothesized that the exploitation of GM fabricated in a controlled manner could be of immense benefit for the sustainable development of effective graphene electrodes with tailorable EC performances.

In this study, we present a strategy to fabricate scalable GM with controllable edge lengths and investigate their edge length-dependent EC reactivity. Specifically, different edge lengths of interconnected GM samples were obtained by exploiting self-cracking TiO_2_ templates with various crack densities. Cyclic voltammetry (CV) was employed to assess EC performance of the fabricated GM and basal graphene samples. The results showed that the electron transfer rate of GM electrodes increased with increasing edge length and was remarkably greater than that of its basal graphene counterpart. This validates the important role of the graphene edge effects in controlling and enhancing EC performance of GM electrodes. To shed light on the possible EC applications of the edge-rich GM electrodes, a high-sensitive glucose detection was conducted on the graphene/Ni hybrid mesh with the longest edge length. The edge-rich graphene/Ni hybrid mesh sensor exhibited a wide linear response range from 10.0 μM to 2.5 mM with a high sensitivity of 1118.9 μA∙mM^−1^∙cm^−2^ and a limit of detection of 1.8 μM. Our findings suggest that edge-rich GM may be a promising candidate for boosting various graphene applications such as graphene-based EC sensors with controlled and enhanced EC reactivity.

## 2. Experimental Section

### 2.1. Fabrication of Ni Mesh Template

The procedure for fabricating the Ni mesh template was adapted from our preceding work [30]. First, the TiO_2_ suspension was prepared by successively mixing the solution of DI water (0.5 mL), ethanol (2 mL) (99%, HY Biocare Chem, New York, NY, USA), acetic acid (4 mL) (99.7%, J.T.Baker, PA, USA), and titanium (IV) ethoxide (0.5 mL) (97%, Sigma-Aldrich, India) under continuous stirring (300 rpm). After stirring 30 min, different volumes (5 µL, 15 µL, 25 µL, 35 µL, and 50 µL) of erasable ink solution (PILOT P-WMRF8-G) were separately added and continuously stirred for 2–3 h. The density of TiO_2_ crack was controlled by amount of added ink. Next, the TiO_2_ suspension (100 µL) was drop-coated onto a 1.5 × 2 cm^2^ cleaned Si substrate with 300 nm thick of SiO_2_ layer atop. Note the SiO_2_ layer is not essential for the formation of TiO_2_ cracks described in below. After a few minutes, an interconnected self-cracked TiO_2_ template was obtained during the evaporation of the solvents in air at room temperature due to particle packing and volume shrinkage [31,32]. Subsequently, Ni (99.99%, Gredmann) was deposited onto the self-cracked TiO_2_ template by a thermal evaporator at a vacuum level of $6\times 10^{-4}$ Pa where Ni slug was heated by a high current (150 A) to melt. The thickness of Ni was controlled by the thickness sensor of the thermal evaporator. Finally, the Ni mesh template was obtained after the lift-off TiO_2_ layer by acetone solution under ultrasonication for 3–5 min.

### 2.2. Fabrication of Graphene Mesh

The graphene was grown on the Ni mesh template with a solid carbon precursor via a rapid thermal annealing approach as reported in previous works [30,33,34,35,36]. The Ni mesh template and cellulose acetate (CA) (C045A047A, Advantec Toyo, Japan) were first put together in the sample container of an infrared lamp annealing system (Mila 5000, Ulvac, Advance Riko, Japan) and then annealed at constant temperature of 850 °C for 8 min at a vacuum level of 0.6 Pa. A heating rate of 10 °C/s was applied to heat from room temperature to 850 °C. After the annealing process, the sample was cooled to room temperature. Subsequently, the GM was obtained after etching Ni mesh by immersing the graphene/Ni hybrid mesh into HCl/FeCl_3_ (0.1 M/0.1 M) overnight. Finally, a gold contact and a SiO_2_ protection layers were successively deposited on top of the GM sample except for the center circular region (with 0.7 cm diameter) using a mask by thermal evaporation and E-beam evaporation methods, respectively.

### 2.3. Characterization

The morphologies and dimensions of the self-cracked TiO_2_, Ni mesh templates, planar graphene and GM samples were examined by optical microscopy (OM, Hamlet MH101, Hwatang, Taiwan), atomic force microscopy (AFM, XE70, Park Systems, Korea), and scanning electron microscopy (SEM, S-3000H, Hitachi Asia Pacific, Taiwan). The crystallographic and layered structures of the GM samples were investigated by Raman spectroscopy (XploRA ONE, Horiba Jobin Yvon, Japan) with the excited laser at the wavelength of 638 nm. The energy-dispersive X-ray spectroscopy (EDS) of the field emission scanning electron microscopy (JSM-6500F, JEOL, Japan) was used to examine the elemental composition of the GM. X-ray photoelectron spectroscopy (XPS) (PHI Hybrid Quantera, ULVAC-PHI, Japan) was used to analyze the surface chemical properties of the graphene samples.

### 2.4. Electrochemical Measurement

Electrochemical characterization was conducted inside a miniature reactor containing a working electrode (GM electrode), a Pt wire counter electrode, and an Ag/AgCl/KCl (3 M KCl) reference electrode (see Appendix A). All the electrodes were connected to a Jiehan 5600 electrochemical workstation. The CV curves were recorded by the electrochemical workstation. The electrolyte consisted of 0.1 M tetrabutylammonium hexafluorophosphate (Bu_4_NPF_6_) (T1279, TCI, Japan) and 1.0 mM ferrocene (87202, Alfa Aesar, China) in acetonitrile (99.5%, J.T.Baker, Pennsylvania, USA)was used to determine the electron transfer rate. Glucose (D-(+) glucose, Sigma Aldrich, Missouri, USA) in 0.1 M NaOH was used to determine the EC sensing capability of the graphene/Ni hybrid mesh electrodes. It has been reported that the current of CVs increase with the pH value of the electrolyte [37]. Since the 0.1 M NaOH solution could yield a high pH value of 13, we chose it as the electrolyte to promote the currents of CVs. The 0.1 M NaOH electrolyte is relevant even for testing the electrode in real sample, e.g., human serum, as reported in [38]. Prior to recording the CV curves, the measurements were repeatedly scanned at the same potential window until no change in the CV curves was observed. All EC measurements were performed by using a cylindrical cell with a diameter of 0.7 cm. Amperometric measurements were carried out by using the electrochemical workstation with a constant potential of 0.6 V and drop-wise addition of glucose solution every 50 s onto 0.1 M NaOH solution without stirring. The volume of added glucose solution was determined by the increased value of glucose concentration. Ascorbic acid (AA) (05878 Fluka, Honeywell, China), dopamine (DA) (Sigma Aldrich, Germany), and uric acid (UA) (Sigma Aldrich, Hungary) were used for the investigation of the interfering species presented in biological samples.

## 3. Results and Discussion

### 3.1. Fabrication and Characterization of Ni Mesh Template

Our fabrication procedure for GM electrode includes (1) drop coating of a TiO_2_ suspension on a SiO_2_/Si substrate to form a TiO_2_ crack template, (2) Ni deposition, (3) lift-off of the TiO_2_ layer, (4) growing graphene on the Ni mesh template, (5) Ni removal, (6) deposition of gold (Au) contact layer and (7) deposition of SiO_2_ protection layer (Appendix A). The gold layer serves to provide a good electrical contact between GM and the potentiostat while the SiO_2_ is a hard protection layer of the graphene electrodes. TiO_2_ suspension was drop-coated onto the Si/SiO_2_ substrate. After few minutes, an interconnected self-cracked TiO_2_ template was spontaneously formed on the substrate due to the particle packing and volume shrinkage properties of TiO_2_ [31,32]. Previous studies reported that crackle pattern, i.e., crack spacing and crack width, can be tuned by changing the thickness of crack layer. However, this approach is complex and may lead the detachment or delamination of the crack layer from the substrate when the crack layer thickness is too thick [39]. Therefore, it is better to find an approach to tune crack spacing without affecting the adhesion of the TiO_2_ layer on the substrate. Herein, we present a facile approach to change crack spacing on the TiO_2_ thin film by adding different volumes of ink solutions into TiO_2_ suspensions with a fixed concentration of other chemicals. Specifically, 5 µL, 15 µL, 25 µL, 35 µL, and 50 µL of ink solutions were separately added to TiO_2_ suspensions to obtain self-cracked TiO_2_ templates (assigned as V-5, V-15, V-25, V-35, and V-50) with different surface coverages (SCs) of cracks, respectively.

Appendix A shows an optical microscopic image of the TiO_2_ template V-50, which clearly illustrates that cracks of the template are well connected to form large amount polygonal cells. A representative crack was characterized by AFM measurement. The width of the crack is around 1.75 μm, as shown in Appendix A. Appendix A displays a SEM image of the Ni mesh sample obtained from the TiO_2_ template V-50 (assigned as NM V-50), and it clearly demonstrates that the Ni wires of the sample NM V-50 are highly interconnected to form a replica of the crack network on the TiO_2_ template V-50. The thickness of all Ni meshes used to synthesize graphene was chosen at 120 nm as indicated by the AFM profile of the NM V-50 shown in Appendix A. From SEM images (Appendix A) of several random positions of the sample NM V-50, the widths of Ni wires are in the 1–3 µm range with an average width of 1.87 µm. The SEM images of five Ni mesh samples (assigned as NM V-5, NM V-15, NM V-25, NM V-35 and NM V-50) obtained from the corresponding self-cracked TiO_2_ templates (V-5, V-15, V-25, V-35, and V-50) are displayed in Figure 1a–e, respectively. These Ni mesh samples exhibit different spacing between Ni wires. A thresholding digital imaging process was used to convert the grayscale SEM images shown in Figure 1a–e to binary images. The surface coverage (SC) of Ni wires of each Ni mesh sample was determined by image processing. Figure 1f shows that the SC of Ni wires of the Ni mesh increases almost linearly with the volume of the ink solution added to the TiO_2_ suspension for the preparation of the TiO_2_ template, which indicates that SC of Ni wires can be easily controlled by the volume of the added ink solution.

### 3.2. Fabrication and Characterization of Graphene Mesh

The graphene was directly grown on the Ni mesh template by a rapid thermal annealing of the solid carbon precursor (CA) at low vacuum. During the annealing procedure, dissociated carbon atoms diffused into the Ni template, and graphene precipitated onto the Ni surface during annealing and cooling [30,33,34,35,36]. After etching off Ni wires, only GM was left on the SiO_2_/Si substrate. As revealed from the SEM image (see Figure 2a) of the GM sample (assigned as GM V-50) obtained from the NM V-50, the GM remained the network distribution like the Ni mesh (NM V-50) template. Figure 2b shows a high magnification SEM image of the GM, which clearly demonstrates that graphene wires are continuous. As indicated by the EDS analysis of the GM V-50 displayed in Figure 2c, there is no presence of Ni element in the GM V-50 sample. We randomly picked several positions of the GM V-50 sample to perform EDS analysis and no nickel element was found at all the positions we examined. This confirms that all Ni wires of the sample were totally removed. The graphitic qualities of all the GM samples, (assigned as GM V-5, GM V-15, GM V-25, GM V-35 and GM V-50) obtained from the corresponding Ni mesh samples (NM V-5, NM V-15, NM V-25, NM V-35 and NM V-50), were characterized by Raman spectroscopy, as shown in Figure 2d. As shown in Figure 2d, all Raman spectra of the GM samples contain the D peak at ~1348 cm^−1^, and two characteristic peaks of graphene; i.e., the G peak at ~1580 cm^−1^; and the 2D peak at ~2704 cm^−1^ [40,41]. The I_2D_/I_G,_ i.e., intensity ratios of the 2D peak over the G peak of all five GM samples are comparable (from 0.69 to 0.82 as shown in Table 1), and it reveals that few layers of graphene were obtained [41]. The presence of the D peak is due to defects or edges of the graphene mesh. Note the Raman spectra shown in Figure 2d were acquired with a fixed laser excitation beam size. Appendix A plots the intensity ratio of the D peak over the G peak (I_D_/I_G_) versus SC of graphene wires of each GM, and it clearly shows that I_D_/I_G_ increases (from 0.31 to 0.88) with SC of graphene wires. It indicates that higher SC of graphene wire can yield more graphene edges.

XPS was used to analyze the surface chemical properties of graphene samples. The C 1s XPS spectra of graphene samples (displayed in Figure 3) can be mainly deconvoluted into four types of carbon bonds: the presence of carbon sp^2^ at 284.5 eV, the presence of carbon sp^3^ (defect peak) at 285.4 eV, and the presence of the C−O−C and O−C=O functional groups at 286.6 eV and 288.9 eV, respectively [42]. Compared to the planar graphene, the ratios of sp^2^/sp^3^ of GM decreased from 7.01 to 4.5 for the GM V-5 and 4.07 for the GM V-50, indicating the presence of high edge in GM samples.

### 3.3. Electron Transfer Rate

CV curves of graphene-based electrodes measured in acetonitrile solution containing 0.1 M Bu_4_NPF_6_ and 1.0 mM ferrocene with different scan rates are displayed in Figure 4a, which clearly shows all of them giving well-defined redox peaks. As the scanning rate increased, the anodic peaks regularly shifted to the right-hand side while the cathodic peaks regularly shifted to the left-hand side. Moreover, their peak currents increased as the scan rate increased which illustrates a quasi-reversible process, indicating effective direct electron transfer between the GM electrodes and the redox species [43,44]. The increase of peak current and the decrease of peak potential separation were observed when the SC of graphene wires increased (see Appendix A). This confirms that the GM electrode with higher SC of graphene wires exhibits better EC reactivity, because it contains more graphene edges to facilitate the EC reactivity, contributed by dangling bonds [14,15,23]. The importance of edge-reactivity can be directly inferred from the comparison of the CV diagram to continuous graphene grown under identical conditions (more information in the Appendix A). Figure 4b presents the linear relationships between the anodic peak positions and the scan rates, as expected for the redox reaction of adsorbed species [45]. Their slopes decreased with the increase of the SC.

Since GM samples were directly grown on the Ni meshes, we assumed that the average width of each GM sample is close to that of Ni wires of each Ni mesh. We have measured the average widths of Ni wires of all five Ni meshes and found their average widths are very close (see Appendix A). Electroactive areas of the electrodes were determined by Randles–Sevcik equation:(1)$$i_{p}=2.69\times 10^{5}AD^{1/2}n^{3/2}\nu^{1/2}C$$
where *i_p_* is the peak current (A), *A* is the electroactive area of the electrode (cm^2^), *n* is the number of electrons participating in the reaction (and is equal to 1), *D* is the diffusion coefficient of ferrocene in 0.1 M Bu_4_NPF_6_ acetonitrile ($D=2.4\times 10^{-5}$ cm^2^∙s^−1^ adapted from [46,47]), *C* is the concentration of the probe molecule in the solution (mol∙cm^−3^), and ν is the scan rate (V∙s^−1^). Table 2 shows the electroactive areas of the GM electrodes determined from Equation (1). In addition, the average edge length of graphene wires of each GM sample was determined from Appendix A. As indicated in Appendix A, the average edge length of graphene wires increases with SC of graphene wires. Consequently, the GM with a higher SC of graphene wire is expected to have a stronger EC reactivity.

The electron transfer rates (*k*) can be extracted from the scan rate dependence of the CV peak positions [48,49]. If the value of the peak separation ∆*E_p_ >* 200/*n* (mV), where *n* is the number of electrons per molecule (*n* = 1 for ferrocene redox mediator), the value of *k* can be calculated from following equation:(2)$$\log k=\alpha \log\left(1-\alpha \right)+\left(1-\alpha \right)\log\alpha -\log\left(\frac{RT}{nF\nu }\right)-\frac{\alpha \left(1-\alpha \right)nF\Delta E_{p}}{2.3RT}$$
where: ν is the scan rate (*V/s*), *R* (= 8.314 J∙mol^−1^∙K^−1^) is the ideal gas constant, *T* is the room temperature (K), *F* is 96485.333 C∙mol^−1^, and α is the transfer coefficient between the electroactive compound and the graphene electrode. α can be determined from plots of scan rate versus peak position $E_{p}=f\left(\log\nu \right)$ which yields two straight lines with one slope equal to 2.3*RT*/(1 − α)*nF* for the anodic peak, and the another equal to −2.3*RT*/*αnF* for the cathodic peak [49] (more information in the Appendix A). According to Equation (2), the electron transfer rate decreases with $\Delta E_{p}$ [43,50]. The *k* value at each electrode was calculated using Equation (2) and plotted as a function of the edge length in Figure 4c. Note that the edge length of each electrode was determined by Appendix A, and the edge length of the planar graphene electrode was assumed to be zero. The *k* value of the planar graphene scattered in ferrocene redox mediator (~0.056 cm∙s^−1^) is close to the results of CVD graphene reported in [45,48]. As displayed, the *k* value increases with the edge length. The red line exhibits a linear fitting of *k* values exhibiting a linear increase of the electron transfer rates from 0.056 to 0.160 cm∙s^−1^ as the edge length of the GM electrodes increased from 0 to 24.04 m. This result confirms that increasing SC of the GM indeed promotes its EC reactivity because of the increase of graphene edge, where more ions from electrolyte are accumulated [14,23].

### 3.4. Application on Enzyme-Free Glucose Detection

For enzyme-free glucose detection, the transition metals (e.g., Ni, Cu, Co) and their oxides are widely used as the metal catalysts due to their high sensitivity, good stability and low-cost [51,52]. Among them, Ni-based materials have received great attention due to their higher sensitivity than other metals which is attributed to the high catalytic activity of NiOOH produced by Ni(OH)_2_ [53]. To illustrate the EC sensing potentiality of the edge-rich GM, a high-sensitivity glucose detection was conducted on the graphene/Ni hybrid mesh (Ni template was not etched). In this investigation, the Ni mesh template, that was used to grow graphene mesh, was utilized as the metal catalyst. For comparison, the glucose detection was performed using the graphene/Ni hybrid mesh V-50 and V-5 samples. Their CV curves processed in 0.1 M NaOH solution at 50 mV∙s^−1^ in the absence and the presence of 0.5 mM glucose are displayed in Appendix A. Both samples showed significantly increased responses in the presence of 0.5 mM glucose in 0.1 M NaOH solution electrolyte. The current intensity obtained from the hybrid mesh V-50 sample was higher than that of the hybrid mesh V-5 sample indicating that the former had a better performance. Therefore, the hybrid mesh V-50 sample was used for further investigation of glucose detection. The SEM image and the EDS analysis of the graphene/Ni hybrid mesh V-50 are displayed in Appendix A. CV measurements processed in 0.1 M NaOH solution at various scan rates are presented in Figure 5a. A pair of redox peaks were observed for all the CV curves at the potential range of 0–0.8 V, which were attributed to the redox process of ${\mathrm{Ni}}^{2+}/{\mathrm{Ni}}^{3+}$. ${\mathrm{Ni}}^{0}$ first reacted with ${\mathrm{OH}}^{-}$ ions to generate ${\mathrm{Ni}}^{2+}$. The ${\mathrm{Ni}}^{2+}$ was then oxidized into ${\mathrm{Ni}}^{3+}$ in the anodic sweep, and the ${\mathrm{Ni}}^{3+}$ was reduced to ${\mathrm{Ni}}^{2+}$ in the cathodic sweep [51]. With the increase of scan rate, the positive and negative shifts were observed for anodic and cathodic peaks, respectively, indicating a quasi-reversible electrochemical process [44]. Figure 5b displays the CVs of the graphene/Ni hybrid mesh V-50 with various glucose concentrations in 0.1 M NaOH solution at a scan rate of 50 mV∙s^−1^. Obviously, when the concentration of glucose increased, a remarkable increase of the current intensity and a slight positive shift of the potential at anodic peak were observed; while the cathodic peak potential was almost stable, suggesting good electrocatalytic activity of the graphene/Ni hybrid material towards glucose oxidation [51]. The reactions can be described as the following equations [51,53]:(3)$$\mathrm{Ni}+2{\mathrm{OH}}^{-}↔\mathrm{Ni}{\left(\mathrm{OH}\right)}_{2}+2e^{-}$$
(4)$$\mathrm{Ni}{\left(\mathrm{OH}\right)}_{2}+2{\mathrm{OH}}^{-}↔\mathrm{NiOOH}+H_{2}O+e^{-}$$
(5)$$\mathrm{NiOOH}+\mathrm{glucose}\to \mathrm{Ni}{\left(\mathrm{OH}\right)}_{2}+\mathrm{glucolacton}$$

The increase of current at both anodic and cathodic peaks in the presence of glucose are similar to the results reported in [53]. We propose that when glucose was added, more NiOOH were needed to react with glucose (see Reaction (5)), which promoted the oxidation process (see Reaction (4)) and the anodic peak current. In the meantime, the adding of glucose also produced more Ni(OH)_2_ through the Reaction (5) and that facilitated the reduction process of $\mathrm{Ni}{\left(\mathrm{OH}\right)}_{2}+2e^{-}\to \mathrm{Ni}+2{\mathrm{OH}}^{-}$ and the cathodic peak current.

Figure 5c displays the typical amperometric response of the graphene/Ni hybrid mesh V-50 electrode with successive additions of glucose solution at 50 s intervals into 0.1 M NaOH solution at applied potential of 0.6 V. As shown, the graphene/Ni hybrid mesh V-50 electrode exhibited a rapid current response after the additions of glucose solution, representing an efficient and rapid response to glucose concentration. Furthermore, a good linear relationship was found between the amperometric current and glucose concentration in the range from 10 μM to 2.5 mM (see Figure 5d), and its regression equation was $i\left(mA\right)=0.373+0.753\cdot C\left(mM\right)$ with $R^{2}=0.9963$. A large intercept was attributed to a large peak intensity due to the high catalytic activity of Ni wires at zero glucose concentration. Based on this equation, the sensitivity of the graphene/Ni mesh V-50 electrode was found to be 1118.9 μA∙mM^−1^∙cm^−2^ obtained by dividing the slope of the regression equation by the active area of the electrode [54]. The limit of detection of the sensor was determined to be 1.8 μM at a signal-to-noise ratio of 3. Although the normal blood glucose level (4.0 to 5.5 mM) [55] is out of the calibration curve range (see Figure 5d) of the graphene/Ni hybrid mesh V-50 electrode, the electrode is still usable for real sample analysis just simply by diluting the real sample in the 0.1 M NaOH electrolyte to have a level of blood glucose within the calibration range. A comparison of our present sensor with other graphene-based non-enzymatic glucose sensors is summarized in Table 3. Its EC analytical performance is comparable to those of the previously reported sensors, and its fast and sensitive response to glucose is attributed to the high electron transfer rate contributed from the edge-rich architecture of the GM and catalysis of Ni material. The selectivity of the graphene/Ni hybrid mesh V-50 electrode was investigated using some interfering species that normally co-exist with glucose in human blood serum, such as AA, DA, and AA [52]. The typical level of glucose in human blood is about 30 times higher than the interfering species [53]. Therefore, the interference experiments were tested by successive additions of 1.0 mM of glucose, 0.1 mM of interfering species, and 0.5 mM of glucose, respectively, into 0.1 M NaOH solution at applied potential of 0.6 V. As displayed in Appendix A, the responses from the glucose are much higher than those from interfering species, exhibiting a good selectivity of the graphene/Ni hybrid mesh V-50 electrode in using as the glucose sensor.

## 4. Conclusions

In conclusion, we have presented a facile and robust approach to fabricate large-scale and high-quality GM samples with controllable edge lengths by the combination of crack lithography and rapid thermal annealing techniques. The edge lengths of interconnected GM samples were controlled by the crack density of TiO_2_ films. The electron transfer rates of the GM samples were evaluated and the result showed an increase in electron transfer rate with increasing graphene edge length. Moreover, we found all the GM electrodes exhibited considerably greater electron transfer rates than that of the basal graphene electrode. This indicates that the abundant graphene edge of the GM electrodes indeed is the main attribution for the enhancement of their EC performance. Furthermore, the sensor made of the graphene/Ni mesh V-50 exhibited a wide linear glucose response range from 10.0 μM to 2.5 mM with a limit of detection of 1.8 μM and a high sensitivity of 1118.9 μA∙mM^−1^∙cm^−2^. The edge-rich GM V-50 presented in this work shows a great promise for advancing various graphene applications such as graphene-based EC sensors with controlled performances.

## Figures and Tables

**Figure 1 nanomaterials-11-00511-f001:**
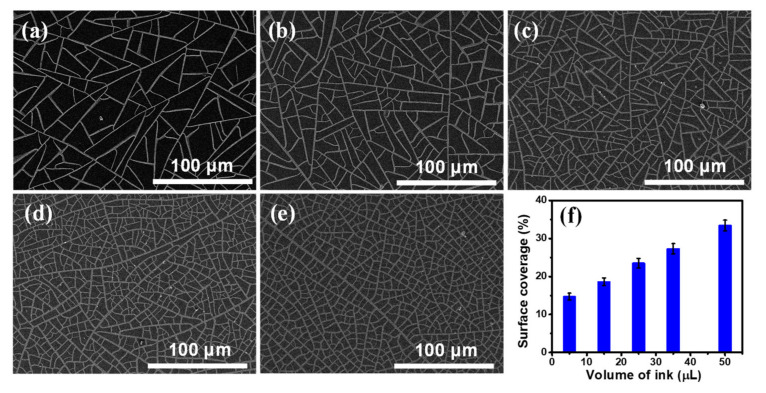
(**a**–**e**) Scanning electron microscope (SEM) images of the Ni meshes: (**a**) NM V-5, (**b**) NM V-15, (**c**) NM V-25, (**d**) NM V-35, and (**e**) NM V-50; (**f**) The surface coverage of Ni wire of each Ni mesh sample versus the volume of the ink solution.

**Figure 2 nanomaterials-11-00511-f002:**
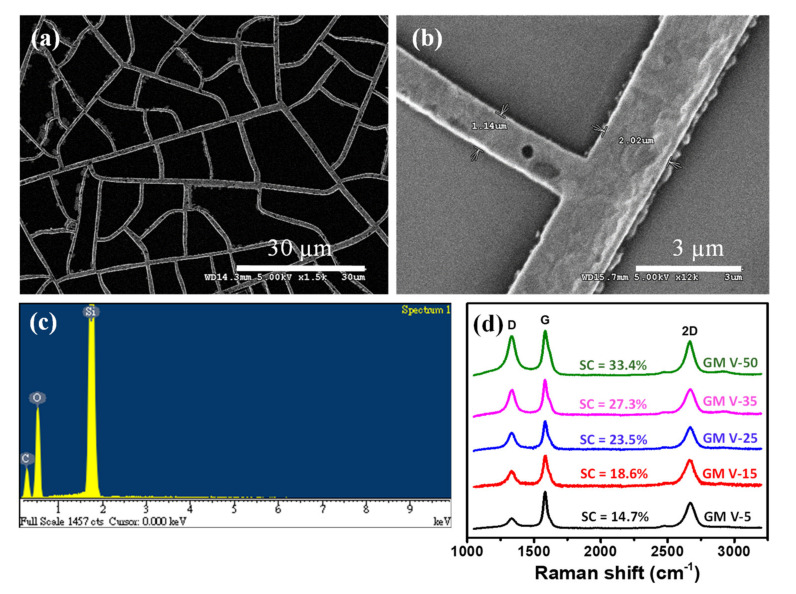
(**a**,**b**) The scanning electron microscope (SEM) image of the GM V-50 sample on a SiO_2_/Si substrate; (**c**) EDS analysis of the GM V-50 on a SiO_2_/Si substrate; (**d**) Raman spectra of GM samples with different SCs of GM.

**Figure 3 nanomaterials-11-00511-f003:**
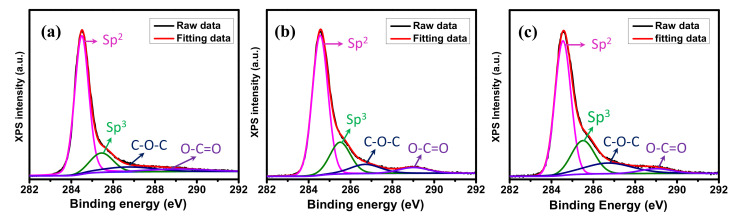
X-ray photoelectron spectroscopy (XPS) C 1s spectra of (**a**) the planar graphene, (**b**) the GM V-5, and (**c**) the GM V-50.

**Figure 4 nanomaterials-11-00511-f004:**
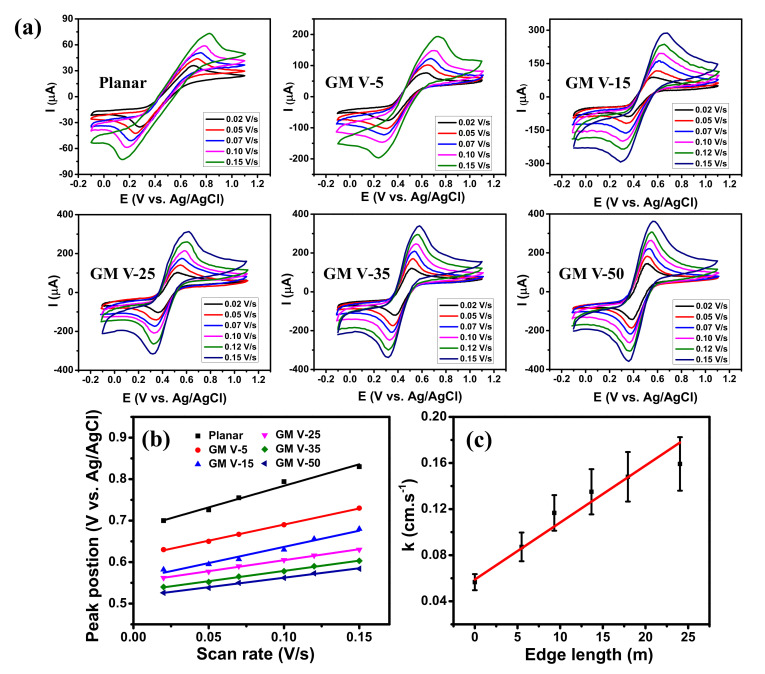
(**a**) CV diagrams of the graphene samples obtained at different scan rates in 1 mM ferrocene and 0.1 M Bu_4_NPF_6_ in acetonitrile electrolyte. (**b**) Anodic peak position vs. scan rate and (**c**) Electron transfer rate constant versus edge length of the graphene electrodes.

**Figure 5 nanomaterials-11-00511-f005:**
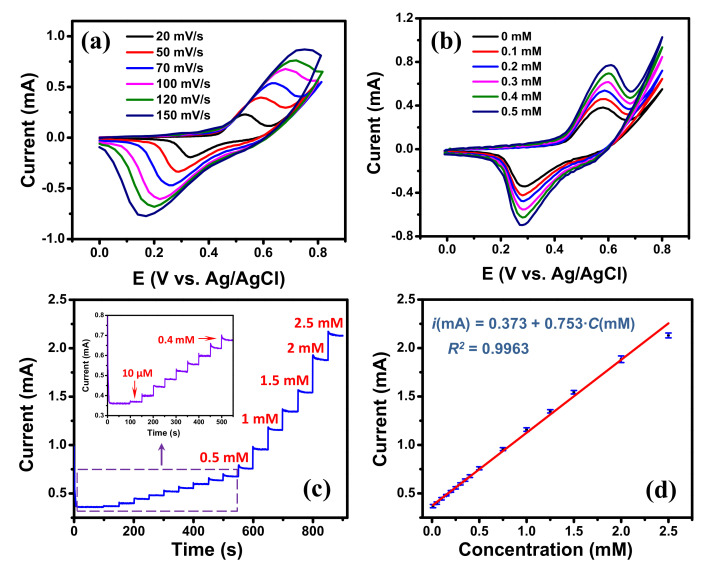
(**a**) Cyclic voltammograms of the graphene/Ni hybrid mesh V-50 electrode at different scan rates in 0.1 M NaOH; (**b**) cyclic voltammograms of the graphene/Ni hybrid mesh V-50 electrode under different glucose concentrations (scan rate: 50 mV∙s^−1^); (**c**) Amperometric responses of the graphene/Ni hybrid mesh V-50 electrode with addition of glucose every 50 s at 0.6 V, (insert show the magnification of low glucose concentration) with scan rate of 50 mV∙s^−1^; (**d**) Linear calibration curve of current response vs. glucose concentration.

**Table 1 nanomaterials-11-00511-t001:** The intensity ratios of the 2D peak over the G peak of all five graphene mesh (GM) samples.

Sample	GM V-5	GM V-15	GM V-25	GM V-35	GM V-50
I_2D_/I_G_	0.69 ± 0.032	0.82 ± 0.04	0.8 ± 0.039	0.73 ± 0.035	0.76 ± 0.03

**Table 2 nanomaterials-11-00511-t002:** The electroactive areas of the graphene mesh (GM) samples.

Sample	GM V-5	GM V-15	GM V-25	GM V-35	GM V-50
Electroactive area (cm^2^)	0.369	0.483	0.538	0.621	0.673

**Table 3 nanomaterials-11-00511-t003:** Comparison of various glucose biosensors. rGO: reduced graphene oxide; GE: graphene; pNi: porous Ni; LSG: laser scribed graphene; NPs: nanoparticles.

Electrode Materials	Linear Range (μM)	Sensitivity (μA∙mM^−1^∙cm^−2^)	Limit of Detection (μM)	Ref.
Graphene/Ni mesh	10–2500	1118.9	1.8	This work
Ni/NiO-rGO	15–6440	1997	1.8	[43]
GE/Ni	10–2500	388.4	0.79	[51]
Cu/Ni-EG/pNi	0.5–1000	6161	0.46	[53]
LSG/Cu-NPs	1–4540	1518	0.35	[54]
Ni nanofoam	5–700	2370	5	[56]

## Data Availability

Data is available upon the reasonable request from the corresponding author.

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
