# Peer review of "Edge-Rich Interconnected Graphene Mesh Electrode with High Electrochemical Reactivity Applicable for Glucose Detection"

_nanomaterials, 2021, doi:10.3390/nano11020511_

Round 1

Reviewer 2 Report

The paper can be published after minor revision reflecting comments inserted as yellow notes into attached pdf of submitted manuscript and supplement - see attached portfolio 1.

Author Response

Thank you very much for your comments. We modified the manuscript and the supplementary material following your comments. Please see the attached PDF for our point-by point responses to the comments.

Reviewer 3 Report

The manuscript "Edge-Rich Interconnected Graphene Mesh Electrode with High Electrochemical Reactivity Applicable for Glucose Detection“ describes the synthesis and characterization of a graphene based electrode of a sensor. For this, a nickel grid was deposited on silicon. The graphene was deposited on top of the grid, and further layers of gold and SiO2 followed. The different steps were monitored using SEM, EDS, Raman spectroscopy and XPS. The electrical response of the full sensor was measured as well. All in all this is a thorough and highly interesting study that deserves publication. I would give way to the publication as it is.

Minor point:

Is the formula 1 correct? I mainly would expect something like: alpha log alpha + (1-alpha)log(1-alpha) - …. Please double check !

I recommend this article as a highlight.

Reviewer 4 Report

In this paper, the authors report a method to produce graphene mesh electrodes with tailorable edge lengths and investigate their edge length-dependent electrochemical reactivity. They show that that the electron transfer rate of graphene mesh electrodes increases with increasing the edge length and is greater than that of basal graphene counterpart.

The authors also investigate the glucose detection sensitivity of the graphene/Ni hybrid mesh.

The topic is timely and interesting. The paper is clear and well-organized. It reports a detailed and thorough study, which is well supported by the experimental data and the literature.

The paper can be accepted after a minor revision:

1. In the introduction: “Graphene has been utilized in various applications such as transparent conducting electrodes [5], energy devices [6, 7], and bio- and chemical-sensing devices [8, 9].” This is quite restrictive. I suggest mentioning at least electronic and optoelectronic devices, such as heterojunctions and transistors (here are a couple of suitable references:  https://doi.org/10.1088/1361-6463/aac562, https://doi.org/10.3390/ma11030345)

2. “In general, EC performance of an electrode is determined by its electron transfer kinetics and the interactions between molecules and electrode surfaces.” Here a reference would be appropriate.

3. “As indicated by the EDS analysis of the GM V50 displayed in Figure 2c, there is no presence of Ni element in the GM V50 sample. This confirms that all Ni wires of the 185 sample were totally removed.” Was EDS analysis performed all across the sample? Is this conclusion based on enough statistics?

4. Figure 5 shows that the current increases linearly with the glucose concentration. I wonder if the process is reversible, i.e. the linearity is kept for decreasing glucose concentration.

Round 2

Reviewer 1 Report

accept in the present form.